stigma; discrimination; healthcare works; COVID-19 management

**Corresponding author:**
U. Venkatesh;
Email: venkatesh2007mbbs@gmail.com

# Factors associated with stigma and manifestations experienced by Indian health care workers involved in COVID-19 management in India: A qualitative study

Ashoo Grover[1], U Venkatesh[2] , Jugal Kishore[3], Tapas Chakma[4], Beena Thomas[5], Geetha Menon[6], Murugesan Periyasamy[7], Ragini Kulkarni[8], Ranjan K Prusty[9], Chitra Venkateswaran[10], Bijaya Mishra[11], Vinoth Balu[12], Maribon Viray[13], Geetu Mathew[14], Asha Ketharam[15], Rakesh Balachandar[16], Prashant Singh[17], Kiran Jakhar[18], Rekha Devi[19], Kalyan Saha[20], Pradeep Barde[21], Rony Moral[22], Ravinder Singh[1], Denny John[23], Jeetendra Yadav[24], Simran Kohli[25], Sumit Aggarwal[26], Vishnu Rao[27] and Samiran Panda[26]

[1]Division of Non-Communicable Diseases, Indian Council of Medical Research (ICMR), New Delhi, India; [2]Department of Community & Family Medicine, All India Institute of Medical Sciences, Gorakhpur, India; [3]Department of Community Medicine, Vardhman Mahavir Medical College & Safdarjung Hospital, New Delhi, India; [4]Division of Non-Communicable Diseases, ICMR-National Institute of Research in Tribal Health, Jabalpur, Madhya Pradesh, India; [5]Department of Social and Behavioural Research, ICMR-National Institute for Research in Tuberculosis, Chennai, India; [6]ICMR-National Institute of Medical Statistics, Ansari Nagar, New Delhi, India; [7]Department of Social and Behavioural Research, ICMR-National Institute for Research in Tuberculosis, Chennai, India; [8]Department of Operational Research, ICMR-National Institute for Research in Reproductive Health, Mumbai, India; [9]Department of Biostatistics, ICMR-National Institute for Research in Reproductive Health, Mumbai, India; [10]Department of Psychiatry, Believers Church Medical College, Tiruvalla, Kerala, India; [11]Department of Clinical Research, ICMR-Regional Medical Research Centre, Bhubaneswar, India; [12]Department of Social and Behavioural Research, ICMR-National Institute for Research in Tuberculosis, Chennai, India; [13]Department of Psychology, Department of Counselling Psychology Martin, Luther Christian University, Shillong, Meghalaya, India; [14]ICMR- Regional Occupational Health Centre -South, National Institute of Occupational Health, Bangalore, India; [15]Division of Clinical Epidemiology, ICMR-National Institute of Occupational Health, Ahmedabad, Gujarat, India; [16]Division of Clinical Epidemiology, ICMR-National Institute of Occupational Health, Ahmedabad, India; [17]Division of Preventive Oncology and Population Health, ICMR-National Institute of Cancer Prevention and Research, Noida, Uttar Pradesh, India; [18]Department of Psychaitry, Government Institute of Medical Sciences, Greater Noida, Uttar Pradesh, India; [19]ICMR-Regional Medical Research Centre, N. E. Region, Dibrugarh, India; [20]Social Sciences and Ethnomedicine, ICMR-National Institute of Research in Tribal Health, Jabalpur, Madhya Pradesh, India; [21]Division of Virology and Zoonotic Diseases, ICMR-National Institute of Research In Tribal Health, Jabalpur, Madhya Pradesh, India; [22]Department of Social and Behavioural Research, ICMR-National Institute for Research in Tuberculosis, Chennai, India; [23]Amrita Institute of Medical Sciences & Research Centre, Kochi, Kerala, India; [24]ICMR-National Institute of Medical Statistics, Ansari Nagar, New Delhi, India; [25]ICMR-National Institute of Medical Statistics, New Delhi, India; [26]Division of Epidemiology and Communicable Diseases, ICMR, New Delhi, India and [27]ICMR-National Institute of Medical Statistics, Ansari Nagar, New Delhi, India

## Abstract

Healthcare personnel who deal with COVID-19 experience stigma. There is a lack of national-level representative qualitative data to study COVID-19-related stigma among healthcare workers in India. The present study explores factors associated with stigma and manifestations experienced by Indian healthcare workers involved in COVID-19 management. We conducted in-depth interviews across 10 centres in India, which were analysed using NVivo software version 12. Thematic and sentiment analysis was performed to gain deep insights into the complex phenomenon by categorising the qualitative data into meaningful and related categories. Healthcare workers (HCW) usually addressed the stigma they encountered when doing their COVID duties under the superordinate theme of stigma. Among them, 77.42% said they had been stigmatised in some way. Analyses revealed seven interrelated themes surrounding stigma among healthcare workers. It can be seen that the majority of the stigma and coping sentiments fall into the mixed category, followed by the negative sentiment category. This study contributes to our understanding of stigma and discrimination in low- and middle-income settings. Our data show that the emergence of fear of the virus has quickly turned into a stigma against healthcare workers.

## Impact statement

This is a study conducted in India to explore the stigma experienced by healthcare workers (HCWs) involved in managing COVID-19. The study aimed to understand the factors associated with stigma and its manifestations, with the goal of formulating intervention strategies to support HCWs.

The study included 93 HCWs from 10 different sites across India. Data was collected through in-depth interviews, which focused on various aspects such as the impact of stigma on their jobs, family life and well-being, coping mechanisms and suggestions for reducing stigma. Thematic analysis was performed on the interview data using NVivo qualitative data analysis software.

The analysis revealed several themes related to stigma among HCWs:

1. Social distance and discrimination: HCWs reported experiencing stigma and discrimination from the public due to their involvement in COVID-19 management. They shared instances where people avoided them, made negative comments or accused them of spreading the disease.
2. Perceived stigma: Some HCWs perceived stigma even though they had not directly experienced it. They mentioned feeling a sense of worry or anxiety among people in their communities or social circles.
3. Neighbourhood stigma: Many HCWs faced stigma from their neighbours and people in their communities. They described instances where neighbours avoided them, refused to talk to them or spread rumours about their work.
4. Avoidance: HCWs expressed feelings of being avoided by known people. They mentioned instances where friends, relatives or even strangers ignored them, which led to a loss of self-esteem.

The study findings highlight the negative impact of stigma on HCWs' lives and well-being. The experiences shared by HCWs indicate the need for intervention strategies to address and reduce stigma. By understanding the factors associated with stigma, policymakers and healthcare organisations can develop support measures and interventions tailored to the needs of HCWs.

## Background

COVID-19 is a worldwide epidemic that has infected 486 million and killed hundreds of thousands of people. India, a country of over a billion people, pronounced inequity and inadequate healthcare systems make combating the pandemic a major challenge. Increased workload, institutional shifts, risk exposure and social stigma, the COVID-19 pandemic considerably impacted the healthcare staff (HCW), especially those involved in COVID-19 screening, tracing and treatment (Collantoni et al., 2021). Stigma is a term that refers to a group of social processes that are used to identify, segregate and discriminate against others in such a manner that limits their (or their group's) life chances and prospects (Link and Hatzenbuehler, 2016). The stigma attached with COVID-19 puts the lives of healthcare workers in jeopardy (Bagcchi, 2020). In the field of healthcare, stigma is a significant impediment to maintaining access to healthcare and ensuring equity and quality. As a result, stigma has a deleterious impact on health outcomes, which is aggravated by mental health outcomes and social isolation, which pose a serious challenge to health services delivery (Sauer et al., 2020). Healthcare personnel who deal with COVID-19 experience stigma and prejudice all across the world, and the stigma attached to COVID-19 puts their lives in danger.

In May 2020, a collective of 13 esteemed medical and humanitarian organisations, including the World Medical Association and the International Committee of the Red Cross, issued a condemnation pertaining to more than 200 instances of COVID-19-related atrocities perpetrated against healthcare professionals. These acts of violence and discrimination can be understood as a manifestation of the interconnectedness between fear propagation, group-level fear contagion, stigmatisation processes targeting individuals perceived as "infectious" and the manifold repercussions they endure, spanning from social exclusion to physical violence (Bagcchi, 2020). During the COVID-19 epidemic, published literature revealed that healthcare personnel, notably front-line doctors, can experience severe psychological stress (Rajkumar, 2020; Uvais et al., 2020). Coping with COVID-19 in healthcare workers and their relatives is a substantial risk factor for psychological distress (Vagni et al., 2020).

During this epidemic, several instances of stigmatisation of healthcare personnel have occurred worldwide. According to media reports in India, doctors and other healthcare personnel working with COVID-19 patients experienced significant societal ostracism; they were evicted from their rented homes and assaulted while performing their duties (Bagcchi, 2020). In India and various other countries, the COVID-19 pandemic has triggered unprecedented fear and anxiety. Unscientific views, beliefs and misunderstandings among the general population are to blame for the COVID-19-associated societal stigma. Furthermore, it has been found that healthcare employees feel themselves to be stigmatised and rejected by others (for example, relatives or friends) because they work in hospitals that treat COVID-19 patients (Uvais et al., 2020). This fast-paced, uncertain atmosphere, as well as traumatic situations, affects not just health professionals, but also their families, friends and co-workers. Fear, mental discomfort, worry, sadness and sleeplessness are all factors that contribute to this psychological crisis among health professionals (Mahmud et al., 2021). Zhu et al. studied health professionals at Tongji Facility in Wuhan, China, which was the designated hospital for the treatment of COVID-19 patients. They discovered that health personnel treating COVID-19 patients had low rates of poor psychological outcomes on the application of psychological preventative measures and relaxation strategies (Zhu et al., 2020).

The stigma among HCWs is seen among those providing care to COVID-19-infected patients, but this also has been observed in various other conditions. It has led to uncomfortable environments for HCWs providing care to HIV-positive patients (Geter et al., 2018), significant hurdles to people's care suspected of being infectious with SARS (Baldassarre et al., 2020), influenza A subtype of H5N1, Middle East respiratory syndrome coronavirus (MERS-CoV), Zika or Ebola (Yuan et al., 2022).

Stigma and discrimination undermine the social fabric, jeopardising the civilisation's ethics and principles to which everyone is entrusted (Baldassarre et al., 2020). However, fear triggered by epidemics should be distinguished from fear based on over-generalised stereotypes.

COVID-19-related mental health problems among HCWs in India have yet to be properly defined. It is very crucial to understand the needs and desires of HCWs when facilities are created and to implement the support measures for them. Qualitative data to study COVID-19-related stigma among HCWs is lacking. There is a paucity of a multi-centric or national-level representative study published on the stigma among healthcare workers in India. The present study aims to explore factors associated with stigma and manifestations experienced by Indian HCWs involved in COVID-19 management. This understanding will enable the formulation, design and implementation of need-based intervention strategies

for healthcare personnel to cope with these psychological and practical challenges.

## Methods

The study was ethically cleared by the National Central Ethical Committee at ICMR vide their letter no NCDIR/BEU/ICMR-CECHR/75/2020, Date: 8th June, 2020; and the Institutional Ethics Committees (IEC) of each site. This analysis was a part of a large multi-centric mixed-method study. There were 10 study sites across India (Figure 1), namely Bhubaneswar (Odisha), Cuttack (Odisha), Mumbai (Maharashtra), Ahmedabad (Gujarat), Noida (Uttar Pradesh), South Delhi, Pathanamthitta (Kerala), Kasaragod (Kerala), Chennai (Tamil Nadu), Jabalpur (Madhya Pradesh), Kamrup (Assam) and East Khasi Hills (Meghalaya).

The data was collected from both public and private facilities from every site, which were chosen purposively based on the feasibility of approaching them. The study group consisted of doctors, nurses, ambulance emergency response teams, lab personnel, X-ray technicians and others who were directly involved in patient care in COVID-designated hospitals, from facilities involved in diagnosis, triaging and referrals services. The healthcare workers chosen for our study included accredited social health activists (ASHA) and community health workers who were involved in case identification, contact tracing, prevention and control measures. The healthcare workers were further stratified as those employed in private and public facilities. The larger study employed a cross-sectional design to calculate the sample size, assuming a 50% prevalence of psychological distress, a 15% non-compliance rate and an alpha error of 5%, with a relative precision of 10%. Quantitative data was collected from a total of 967 participants across the 10 states. To gain a deeper understanding of the stigma and its manifestations, approximately one-tenth of this sample size, which is approximately 93 participants, was included for qualitative investigation. The qualitative investigation covered a total of 93 participants across various sites, with 9–10 respondents per site. This sample size was considered sufficient to reach data saturation, which is crucial for comprehensive qualitative data analysis.

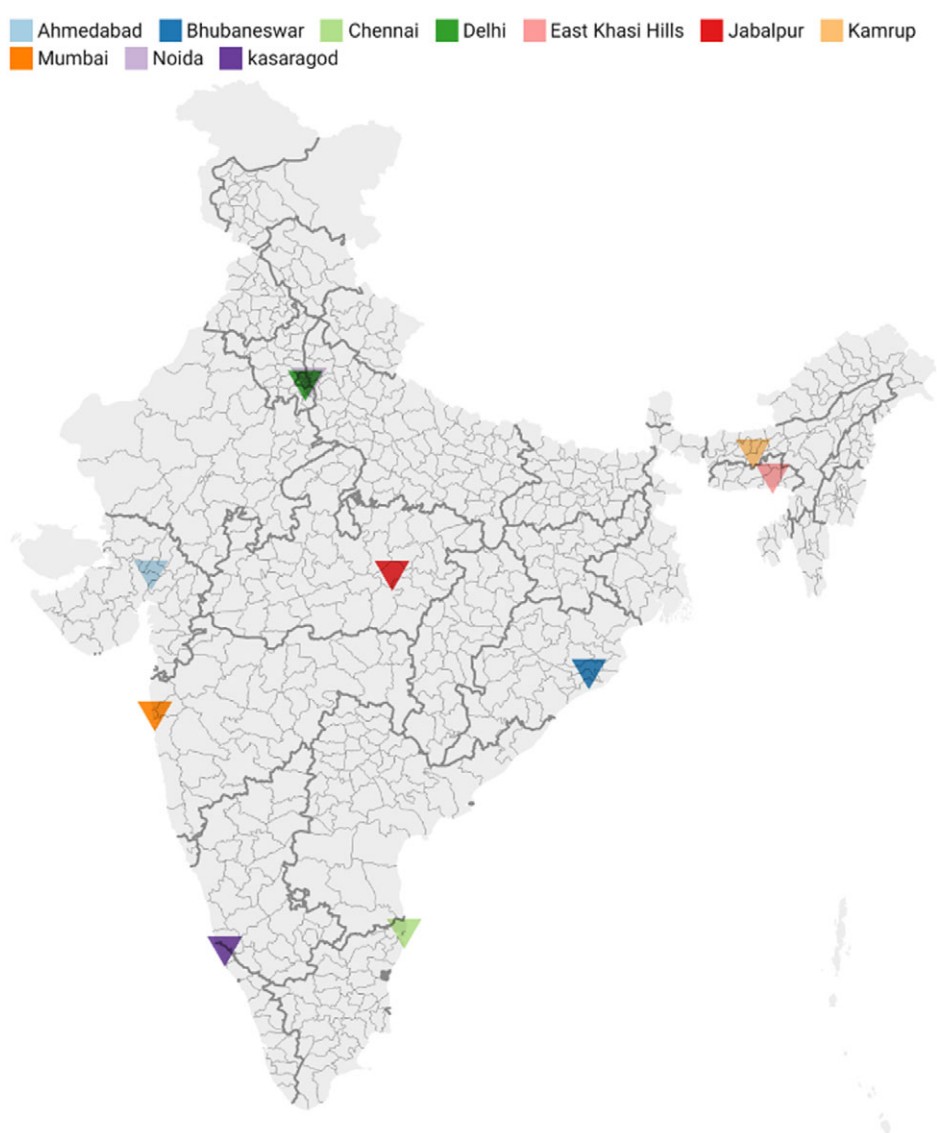

**Figure 1.** Spatial distribution of sites participating in the study.

**In-depth interview:** Based on the mutual convenience of the investigator and the participants, an appointment was arranged for a telephonic interview with 93 participants across 10 centres during October–December 2020. Consent forms and information sheets were then read out. An audio consent was obtained from the participants. Their consent was also acquired for the interview to be audio-recorded and also for the transcripts to be used in the future. The participants were also informed that they had the option to leave the study at any moment. After that, each interview was digitally recorded, and notes were taken at the same time. To protect the participants' safety and privacy, any personal identifying information disclosed during the interview was encrypted. All interviews were translated from the local language to English and transcribed into a Word document. The interview focused on five primary areas: impact on their job, influence on family life, impact on the sense of well-being, coping with COVID-19 and suggestions for reducing the stigma associated with COVID-19. Each domain was given a set of questions to answer, followed by context-relevant probes to elucidate and unpack their responses, experiences and perceptions with respect to the phenomena under study (Supplementary Table 1).

**Thematic analysis:** NVivo qualitative data analysis software, Version 12 (QSR International Pty Ltd., London, UK, 2020) was used to analyse the data. The site investigators reviewed the audio-recorded interviews before the analysis was conducted at the coordinating centre. The data was imported into the NVivo software package to link the concepts contained in the data. To gain deep insights into the intricate phenomenon, thematic analysis was performed by categorising the qualitative data into meaningful and related categories. The researcher (1) became familiarised with the content and information in the transcriptions; (2) read each transcription line by line to find keywords, meanings or concepts related to the study's objectives; (3) refined coding by assembling similar codes and attempted to group codes together; (4) collected all texts coded with the same code in one place using the Microsoft Word programme; (5) organised and gathered all coded data into themes and (6) common themes were identified and inferences were drawn from them, with sample quotes chosen to illustrate the emerging themes.

**Sentiment analysis:** NVivo's sentiment scoring employs a scoring system. Each sentiment-containing word has a pre-determined score. Each sentiment node represents a range on a scale (of sentiment), ranging from very negative to very positive. Using the NVivo package "sentiment," we calculated and aggregated text polarity sentiment at the coronavirus heading level and specific words of the phrases for insightful findings. The sentiment analysis software is based on a database of positive and negative words and phrases. (Figure 2).

## Results

The study included a total of 93 healthcare participants, of which 53.7% were females, and 46.2% were male. Figure 3 depicts the site-specific distribution of healthcare workers who participated in our study.

### *Thematic analysis*

Healthcare workers usually addressed the stigma they encountered when doing their COVID duties under the superordinate theme of stigma. Among them, 77.42% said they had been stigmatised in some way. The stigma is coded into subthemes; Figure 4 shows stigma themes clustered by word similarity. The analyses revealed seven interrelated themes surrounding stigma among HCWs.

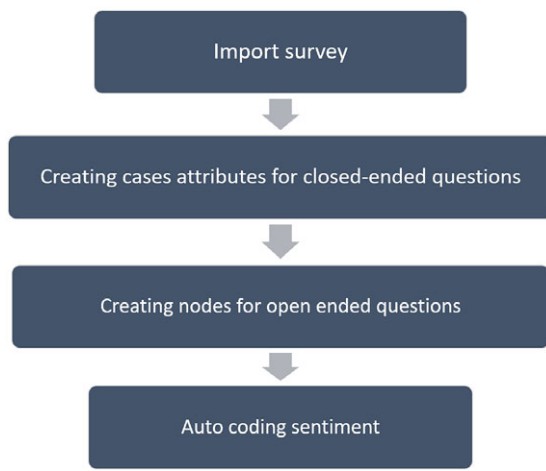

**Figure 2.** Steps used during sentiment analysis done using NVivo software.

a.  Social distance and discrimination

In the battle against COVID-19, the word "social distancing" is often used. It is recommended to maintain social distance. In fact, there is a physical distance of approximately 6 feet between two people. On the contrary, the use of the term social distancing has promoted social stigma, which has led to nearly an attitude of boycotting COVID-19-related individuals. Most of the study participants experienced stigma and discrimination against HCWs involved in COVID-19 management and expressed a range of emotions in response to stigma, as described below by participants.

*"When I used to go back to my house, everyone used to run away from me. Nobody was ready to even sit close to me." – House Keeping staff, 42 years old, Male, Noida*

*"One day, I was doing a fever camp and I completed it around 10:30 am, and then I was supposed to be at the second camp by 11:30 am. In between, there was a complaint from a woman who was waiting for me. That woman was really angry and started blasting me for a slight delay. She then accused me of coming from a patient's home and held me responsible for spreading the disease." – Doctor, 26 years old, Female, Chennai*

*"When people see me, they say 'Here comes Corona!'. They (neighbours) didn't behave well. What can we do about it? They would close their doors and windows if they saw me coming from a distance. At marketplaces, people cover their faces when they see me. They would cross me or supersede me very fast by accelerating their vehicles. I even avoided my relatives completely. When I went to them, they went away. They didn't want to be face-to-face with me. There was hatred in their minds. They would avoid me and would go away on seeing me." – Ambulance driver, 55 years old, Bhubaneswar*

b.  Perceived stigma

Perceived stigma is entirely subjective, indicating how people perceive themselves to be stigmatised and believe they are subjected to discriminatory conduct by others, independent of the real stigmatic ideas and actions. During the interview, we found that a set of HCWs strongly perceive stigma although they have not experienced it.

*"They all asked whether I drive a vehicle with Covid patients. There was a function with 20 members near my house. It was an engagement function. Even though they had invited me, I feel that they*

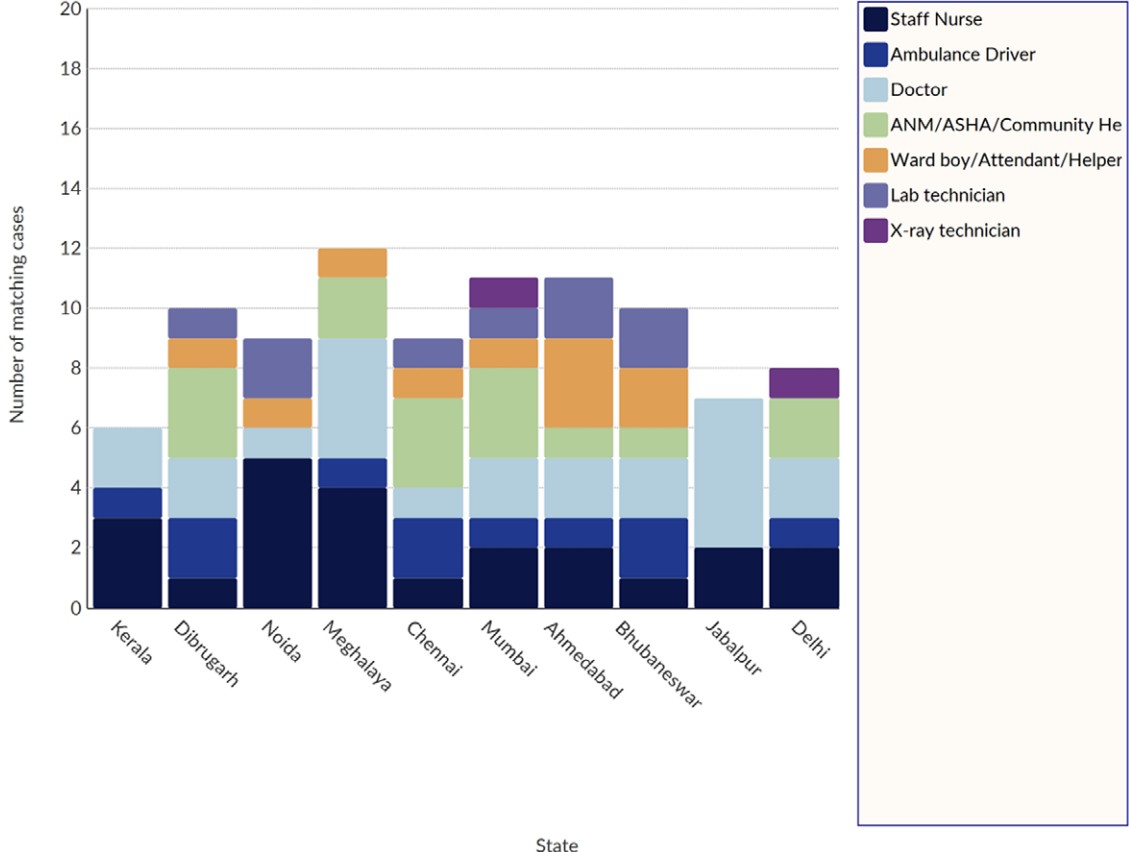

**Figure 3.** Site-wise distribution of healthcare workers (HWCs) participating in our study.

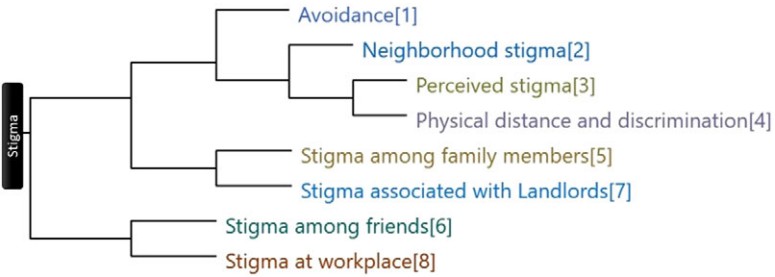

**Figure 4.** Horizontal dendrogram showing stigma themes clustered by word similarity.

*would have worried thoughts in their minds. So, I didn't attend that function so as to avoid unnecessary talks and complaints about me. I avoid going to most of the functions near my house, like engagements and birthday parties. Worry will be there in their minds as I am driving a vehicle with Corona patients."* – Ambulance driver, 30 years old, Male, Kerala

*"I cannot say that I have experienced something like that. During the initial COVID-19 period, people have expressed huge hatred for the disease. The local people near the sub-centre where I live had a fear or anxiety within them, especially during the initial days of the outbreak, but they have not expressed any highly intolerant hateful behaviour towards COVID-19."* – ASHA worker, 42 years old, Female, Dibrugarh

*"Ah, sometimes it's evident from their body language, their actions, or at times, even from the individuals who don't turn up when we*

*expect. I remember when we had our first patient, even my staff stood at a distance."* - Doctor, 35 years old, Male, Meghalaya

c. Neighbourhood stigma

The majority of people experienced stigma from their neighbourhood at some point in time. They faced this experience while coming home after performing duties such as going to the market, shopping, etc. Some of these participants experienced stigma in the neighbourhood where they had grown up or spent most of their lives. For example,

*"My neighbours used to think that if I am working in the COVID-19 ward, I am COVID-19 positive myself. So, they stopped talking to me and my family. They used to hide inside on seeing me on the road. They even used to tell the vegetable and fruits vendors that she is working with Corona patients, so the vendors refused to sell me vegetables."* – Nurse, 33 years old, Female, Jabalpur

*"When we went to the village and when I used to say that I work in a hospital, people there avoided coming close to me. They were not even ready to talk to me. When I was quarantined in my house, my other brothers and friends also were not ready to come near me or meet me."* –X-ray technician, 56 years old, Male, Maharashtra

*"After 10 days of Covid-19 duty, I used to get 7 days to quarantine. During those 7 days, once I was passing through my neighbourhood to buy bananas, when I heard my neighbour saying, 'Look at her, she is roaming around being in quarantine'. I even get worried about entering shops with my ID card, thinking it would worry the shopkeepers."* – Nurse, 47 years old, Female, Kerala

*"My neighbours and relatives kept a distance from me. Even after my report tested negative and I stayed isolated for three to four days, they said, 'Brother, you stay away from us.' If my car would get damaged, then nobody would give me a lift also."*– Nurse, 28 years old, Male, Noida

d. Avoidance

Avoidance-related stigmatising thoughts are connected with considerable loss of self-esteem. Some participants expressed the feelings that known people are avoiding them as described below.

*"Once I and my sister, who is also working in a hospital, were under a home quarantine routine, people often ignored us despite clear eye contact. It hurt us because we never knew their reason to avoid, but we consoled ourselves to their ignorance with times."* Nurse, 27 years old, Female, Dibrugarh

*"I mean, what happened was that those people who know about it, who understand this scenario, have fewer people in that category, as in 5–10% people in that context, despite knowing me. When they come to the hospital, their behaviour would be different, but if they remain around us, near us, then their behaviour would change, Or they keep some distance from you, something that we can call paranoia."* -Doctor, 32 years old, Male, Noida

*"It really hurts, they talk so much. I could come back home every day after work, as we didn't come in direct contact with the patients and overall we were following protocols, so we were allowed to come home; we just had to transport the patients, but people take it otherwise, that is why there is this stigma in the society. It also affected my sister, but she is very brave, and she cares about me a lot. But even then, because of this, I didn't stay at home anymore even though I could. I preferred staying in a hotel instead."*- Ambulance driver, 34 years old, Female, Meghalaya

e. Stigma at work place

Stigma in the workplace can influence people's attitudes and views about persons who are working in COVID care. A few proportion of people experienced this and shared their feelings. For example.

*"Whenever I encounter stigmatic experiences, I feel people are just using us; they didn't bother me even when I am in a problematic situation. I have recently been posted in another district for COVID duty. The same department staff was scared to talk to me or come close to me since I came from Chennai. It makes me feel stressed."*- Nurse, 32 years old, Female, Chennai

*"Some staff at my workplace say that you're working in a Covid ward and do not need to come near like that, wear the mask properly, and whatnot, like that. They used to say don't come since you are working there; you might bring the disease and even get infected."* -Nurse, 32 years old, Female, Meghalaya

f. Stigma associated with landlords

Few HCW landlords forced HCW to vacate the facility on multiple occasions through various means because they were engaged in COVID-19 duties. They felt that the friendly neighbourhoods are turning into frigid, stigmatised zones of irrationality and bias. For example.

*"Our landlord started mentally mistreating us for staying in their house during the period just because we are COVID staff. People are afraid of us during door-to-door survey work in villages, even if we're in PPE for the whole day. We are also not provided with govt. quarters while we were working with COVID."* – Community Health Worker, 29 years old, Male, Bhubaneshwar

*"Actually, when I used to stay with my landlord, I had to face a lot of issues, and it was really difficult for me to continue my stay there because I couldn't change their mentality, and I also couldn't stop working because of them. So, I discussed this in my hospital, and they allowed me to stay at a hotel. I used to stay in the hotel with the same people who were on COVID duty."* – Doctor, 27 years old

*"I can say that there's a lot of stigmatisation on us, for example, if those who are staying in rented houses are healthcare workers, landlords don't want to rent them anymore. We cannot go for a haircut because we are healthcare workers, and to be honest, sometimes we don't get to do the everyday things we want to do because of stigmatisation."* – Nurse, 34 years old, Female, Meghalaya

g. Stigma among family members and friends

They were few proportions of participants who felt stigma from close friends or family members. For example.

"I mean, of course they are caring, but it is just that you know, like when your family members are scared of you, it is not a nice feeling. Especially when you feel accomplished at work, when you feel good at work, you have done something good and then come home at work, they appreciate you, but the home they are scared of you. When I am going home, they all will prepare not to stand in the way……in the workplace, we are okay because we are all on the same boat" -Doctor, 32 years old, Female, Meghalaya

*"My friends stopped coming to me, and they stopped meeting me completely. So, everything had changed suddenly. Everything in my life is not the same and normal as before, and a lot has changed still. Since I have started doing the duty related to COVID, all this, I mean everything, has changed. Nobody has got anything to do with me, neither do I go to them."*- Nurse, 26 years old, Male, Noida

### *Sentiment analysis*

The sentiment analysis of the stigma and the coping topic is shown in Figure 5. The red colour represents negative sentiments, and dark green shows positive, while the orange colour represents mixed and grey represents neutral sentiments. It can be seen that the majority of the stigma and coping sentiments fall into the mixed category, followed by the negative sentiment category.

### Discussion

This study contributes to our understanding of discrimination in LMIC settings and the contextual factors that influence it in India. Our interview data show the emergence of COVID-19-related stigma; fear of the virus quickly turned into fear of discrimination

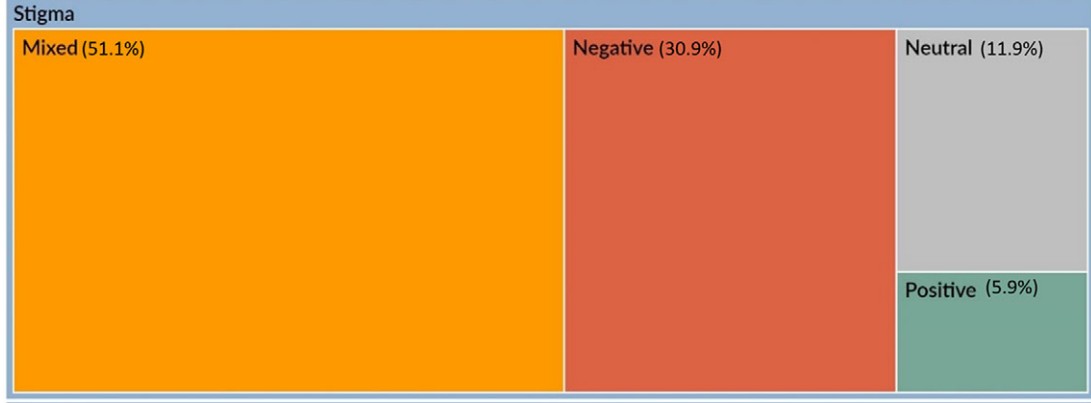

**Figure 5.** Hierarchy chart showing nodes compared by percentage of sentiment coding references.

against HCWs because HCWs are involved in the care of COVID-19 patients. Similar findings were found during the early stages of the pandemic in India as well as globally, where a relatively high level of stigma associated with COVID-19 was observed among HCWs (Jeon and Kim, 2022; Zandifar et al., 2020; Jain et al., 2021; Yufika et al., 2022). Our evidence supported by evidence found by that some stigma may reflect one's desire to "keep people away" by Phelan et al. (2008). Stigma has not been discussed when one HCW discriminates against the other. In our study, we discovered cases of stigma and discrimination experienced by HCWs at the hands of other HCWs. Grover et al. have also observed similar instances of stigma and discrimination in workplaces of HCWs (Grover et al., 2020). Our few study participants faced stigma from close friends or family members, such as when your family members are scared of you.

Giri et al. (2022) revealed that healthcare workers in Nepal experienced significant levels of perceived stress and stigma during the COVID-19 pandemic, while also highlighting the crucial role of social support in mitigating these negative effects (Giri et al., 2022).

The negative effects on front-line HCWs may be more severe when the group receives more attention in the press and public media. Cuong Do Duy et al. demonstrated a higher level of stigma in "negative self-image" and "concerns about public attitudes" domains. The most common was a feeling of guilt towards family members and friends, as well as a desire to avoid contact with neighbours and the community (Do D Duy et al., 2020).

Similarly, findings from Narita et al. (2023) suggest that COVID-19-related discrimination contributes to the development of post-traumatic stress disorder (PTSD) symptoms and psychological distress among healthcare workers, highlighting the urgent need for interventions to address stigma and support the well-being of healthcare workers in the context of COVID-19 management (Narita et al., 2023).

Our study also found stigma faced by HCWs who are living in rental houses and apartments. They had been given eviction notifications and were publicly disgraced with discriminating mindsets. A similar finding is also reported by R. V. Radhakrishnan et al., where HCWs were forced to vacate the facility on multiple occasions through various means because they were engaged in COVID-19 duties (Radhakrishnan et al., 2021). It has been known that healthcare workers involved in COVID-19 management faced COVID-19-related stigma, which was associated with various psychological challenges (Gaber et al., 2023).

Sachdeva et al. (2022) concluded in their study that HCWs caring for COVID-19 patients experienced and reported significantly high levels of stigma, particularly with regard to disclosure issues and concerns about the public's perception (Sachdeva et al., 2022). Similarly, a cross-sectional survey done by Ranjit et al., 2022 of 150 healthcare workers in India showed that almost 50% experienced discrimination due to their association with COVID-19 patients, highlighting the prevalence and impact of courtesy stigma on the mental health of healthcare workers (Elsaid et al., 2022). We also found that during the COVID-19 pandemic, workplace violence against healthcare workers was prevalent, with more than 60% of participants reporting exposure. Verbal violence was the most common type of experienced incidents, while age, gender and work specialty were found to be associated factors (Ranjit et al., 2022).

Some kinds of stigmatising attitudes shown by the society were hurtful social distancing, avoidance, rejection and refusal. Figure 6 shows factors associated with stigma that aggravated the manifestation of HCWs derived from our study findings.

HCWs report significant changes in their work-life situation, including high workload with inconsistent timings compounded by longer periods of cumbersome personal protective equipment usage, periods of quarantine and long periods of separation from family. The biggest obstacle was being apart from family, the challenge of caregiving, especially for females with infants and children, and the anxiety of infecting relatives. The fear of contagion fuelled stigma from the community and peers, which appeared as avoidance and rejection. Coping mechanisms included friend and family support, as well as positive experiences such as praise and acknowledgement for their role in the pandemic (Chakma et al., 2021).

Support from family, peers and co-workers helped them in coping with the COVID-19 pandemic. In order to mitigate the stigma, all HCWs suggested educating people on COVID-19. Proper public health education is required to avoid social harassment of healthcare personnel. Media, organisations and the government should work together to reduce and mitigate stigma. To tackle the social stigma caused by COVID-19, creating an environment where open dialogue among people and HCWs is possible is required. The once-honoured white coat has now been labelled as a sign of diseased and unclean things. Furthermore, individuals who have recovered from COVID-19 are also discriminated against, and not just active COVID-19 patients and healthcare practitioners.

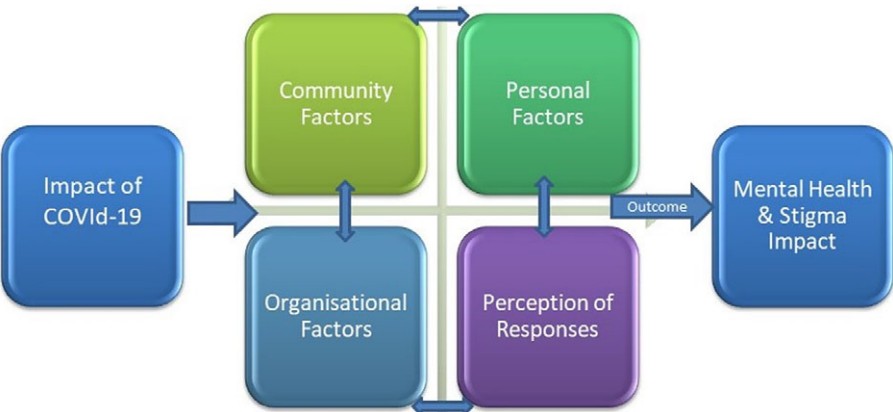

**Figure 6.** Factors associated with stigma derived from our study findings.

The outcomes of our study will help healthcare practitioners comprehend the stigma they face as a result of COVID management at various levels.

This research could aid in the development of feasible, timely and need-based interventions for healthcare professionals that address these concerns. The treatments based on these findings will safeguard the interests of healthcare workers and give the medical, healthcare system, and, most importantly, psychosocial and emotional support they require to deal with the management of this difficult illness. Need-based interventions will represent holistic COVID care that will make a difference in reducing COVID-19-related morbidity and mortality by promoting COVID-19 screening, testing and treatment. In case of future epidemic/pandemic situations in the country, the findings of the study will be significant for managing and developing strategies to alleviate mental health concerns among healthcare staff.

## Conclusion

This study presents an attempt towards understanding the perceived stigma and discrimination faced by healthcare workers during the COVID-19 pandemic. In the study which included a total of 93 healthcare participants, 53.7% were females, and 46.2% were male.

In-depth interviews with the healthcare workers followed by a thematic analysis of the contents led to the revelation of seven interrelated themes surrounding stigma among HCWs, including Social distancing and discrimination, Perceived stigma, Neighbourhood stigma, Avoidance-related stigma, Stigma at workplaces, Stigma associated with landlords and Stigma faced at the hands of family and friends.

The sentiment analysis of the stigma and the coping topic was performed using the NVivo's sentiment scoring system in which each sentiment node represents a range on a scale (of sentiment), ranging from very negative to very positive. The study concluded that the majority of the stigma and coping sentiments fall into the mixed category, followed by the negative sentiment category.

## List of abbreviations

| | |
|---|---|
| **HCWs** | healthcare workers |
| **PPE** | personal protective equipment |
| **LMIC** | low- and middle-income countries |
| **ASHA** | accredited social health activist |
| **SARS** | severe acute respiratory syndrome |
| **HIV** | human immunodeficiency virus |

**Open peer review.** To view the open peer review materials for this article, please visit http://doi.org/10.1017/gmh.2023.40.

**Supplementary material.** The supplementary material for this article can be found at http://doi.org/10.1017/gmh.2023.40.

**Data availability statement.** Available data and material.

**Acknowledgements.** Delighted to appreciate all subjects who participated in the study or withdrawal.

**Author contribution.** Study conception and design A.G., U.V., J.K., T.C., B.T., G.M., M.P., R.K. Data collection, C.V., V.B., M.V. Data analysis and interpretation, A.K., R.B., P.S.
Drafting of the article A.G., U.V. and J.K.
Critical revision of the article A.G., U.V. and J.K. The authors read and approved the final manuscript.
**Study conception and design by** A.G., U.V., J.K., T.C.

**Financial support.** This study was funded by the ICMR, New Delhi, through the National Task Force for COVID-19.

**Competing interest.** The authors declare that they have no competing interests.

**Ethics standard.** An informed permission was taken from the National Central Ethical Committee at ICMR vide their letter no NCDIR/BEU/ICMR-CECHR/75/2020, Date: 8th June, 2020; and the Institutional Ethics Committees (IEC) of each site.

**Consent for publication.** Oral consent from the study subjects had been obtained for publication.

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
