## [Reviewer Report]

Respected Sir/Madam, 

We wish to submit an original research article entitled “Stigma drivers and manifestations experienced by Indian Health Care Workers involved in COVID-19 management in India: A Qualitative study” for consideration by your renowned Journal. We confirm that this work is original and has not been published elsewhere, nor is it currently under consideration for publication elsewhere.

In this manuscript, we have reported that Healthcare workers (HCW) usually addressed the Stigma they encountered when doing their COVID duties under the superordinate theme of Stigma. Among them, 77.42 % said they had been stigmatized in any way. Analyses revealed seven interrelated themes surrounding Stigma among health-care workers. It can be seen that the majority of the Stigma and coping sentiments fall into the mixed category, followed by the negative sentiment category.

This study contributes to our understanding of stigma and discrimination in low- and middle-income settings. Our data show that the emergence of fear of the virus has quickly turned into a stigma against Health-care workers.

We believe that this manuscript is appropriate for publication by your renowned journal because it specifically aims at public and mental health problem and concept of Cambridge Prisms: Global Mental Health which are among one of the main objectives of your journal.

In this study we conducted in-depth interview across ten centers in India which was analyzed using NVivo software version 12. Thematic and sentiment analysis was performed to gain deep insights into the complex phenomenon by categorizing the qualitative data into meaningful and related categories.

Cambridge Prisms: Global Mental Health being a pioneer journal in field of Mental health and Medicine is a perfect place for this type of work as the readership of this journal is quiet interested in viewing research work which would address the areas that are untouched or have very few literature available.

We have no conflicts of interest to disclose.

---

## [Reviewer Report]

This paper presents the results of a qualitative study examining the experiences of stigma faced by 93 healthcare workers from 10 distinct centers in India in relation to the COVID-19 pandemic.

As there are few studies of this type and methodology covering this population in the given region, the results presented here are important from a public health policy perspective.

Certain aspects of this paper require correction or clarification, and these are enumerated below:

1. The term “stigma drivers” is not a standard one used in the literature. Moreover, as this is a qualitative study, direct inferences regarding causality or mediation cannot be drawn. It would be better to use a neutral term such as “Factors associated with stigma” in the title as well as the text.

2. In the introduction, the authors refer to “200 COVID-19 related atrocities”. However, a careful reading of the original paper cited by them suggests that this term is somewhat exaggerated. The incidents described in the original paper cover a wide range of experiences, from subtle forms of discrimination / avoidance to actual acts of physical violence. Moreover, these are related not so much to “COVID-19” as to stigma, fear of infection, fear contagion, etc. This sentence should be reworded more carefully to underline the links between fear of COVID-19, fear contagion in groups / communities, stigmatization of those perceived to be “infectious” (including healthcare workers) and the wide range of problems they may face as a result, ranging from exclusion / avoidance to actual “atrocities” (e.g., physical violence).

3. The Introduction is somewhat limited in its scope as it focuses mainly on older literature from the period 2020-2021. Recent research in this field, particularly from the same geographical region, could be cited. Some studies worth mentioning in this regard are: Giri et al. (2022), Narita et al. (2023), Gaber et al. (2023), and Sachdeva et al. (2022) - the latter involving Indian healthcare workers. This literature should be cited and discussed where appropriate both in the Introduction and in the Discussion.

4. Given the authors' concerns about including a wide range of healthcare workers, the authors should provide more details about the sampling method / sample size estimation. Was a representative number of staff from each category (doctors, nurses, technicians, support staff, etc.) recruited from each center? This should be mentioned in the Methodology. Any barriers to recruitment / frequent refusals and the reasons for the same could also be mentioned.

5. Concerning the Discussion, see point #3 above. The authors could also discuss more specialized studies, such as those covering violence against healthcare workers during the pandemic (Elsaid et al., 2022) or other Indian studies that have been published in 2022-2023 (e.g., Ranjit et al., 2022).

6. Many of the authors of this study appear to have conducted a similar study in 2021 (https://doi.org/10.4103/ijmr.ijmr_2204_21); it would be useful to compare the results they have obtained there with those of the current study.

7. Figures 2 and 7 are not very informative and may either be deleted, or submitted as supplementary material.

---

## [Reviewer Report]

Dear [Editor’s Name],

I hope this letter finds you well. I am writing to submit the revised version of our article entitled “Factors associated with stigma and manifestations experienced by Indian Health Care Workers involved in COVID-19 management in India: A Qualitative Study” for consideration in “Cambridge Prisms: Global Mental Health.” We greatly appreciate the opportunity to revise and resubmit our work following the constructive feedback provided by the reviewers.

We are grateful for the valuable feedback and suggestions provided by the reviewers, which have undoubtedly strengthened the quality and contribution of our research. We have carefully addressed each point raised, and have made significant revisions to improve the clarity, organization, and overall impact of our article.

We believe that the revisions have significantly improved the quality and contribution of our article and have addressed the concerns raised by the reviewers. We remain confident that our work aligns with the scope and objectives of “Cambridge Prisms: Global Mental Health” and will be of interest to its readership.

Thank you once again for considering our revised submission. We are grateful for the opportunity to contribute to the field of global mental health through the esteemed platform provided by “Cambridge Prisms.” We remain committed to working closely with the editorial team to ensure the successful publication of our article.

Please find attached the revised version of our manuscript in Microsoft Word format, along with a point-by-point response to the reviewers' comments.

Sincerely,

Dr U Venkatesh

Assistant Professor

Department of Community Medicine & Family Medicine

AIIMS Gorakhpur

---

## [Reviewer Report]

The revisions made by the authors are satisfactory in my opinion. I have no further major changes or corrections to suggest.